# Capturing Transitional Pluripotency through Proline Metabolism

**DOI:** 10.3390/cells11142125

**Published:** 2022-07-06

**Authors:** Gabriella Minchiotti, Cristina D’Aniello, Annalisa Fico, Dario De Cesare, Eduardo Jorge Patriarca

**Affiliations:** Stem Cell Fate Laboratory, Institute of Genetics and Biophysics, A. Buzzati-Traverso, CNR, 80131 Naples, Italy; cristina.daniello@igb.cnr.it (C.D.); annalisa.fico@igb.cnr.it (A.F.); dario.decesare@igb.cnr.it (D.D.C.)

**Keywords:** naïve-to-primed pluripotency, proline, proline metabolism, metabolic reprogramming, amino acid stress response pathway, collagen hydroxylation, histone hydroxylation, DNA methylation, primordial germ-like cells, gastruloid competence

## Abstract

In this paper, we summarize the current knowledge of the role of proline metabolism in the control of the identity of Embryonic Stem Cells (ESCs). An imbalance in proline metabolism shifts mouse ESCs toward a stable naïve-to-primed intermediate state of pluripotency. Proline-induced cells (PiCs), also named primitive ectoderm-like cells (EPLs), are phenotypically metastable, a trait linked to a rapid and reversible relocalization of E-cadherin from the plasma membrane to intracellular membrane compartments. The ESC-to-PiC transition relies on the activation of Erk and Tgfβ/Activin signaling pathways and is associated with extensive remodeling of the transcriptome, metabolome and epigenome. PiCs maintain several properties of naïve pluripotency (teratoma formation, blastocyst colonization and 3D gastruloid development) and acquire a few traits of primed cells (flat-shaped colony morphology, aerobic glycolysis metabolism and competence for primordial germ cell fate). Overall, the molecular and phenotypic features of PiCs resemble those of an early-primed state of pluripotency, providing a robust model to study the role of metabolic perturbations in pluripotency and cell fate decisions.

## 1. Introduction

Pluripotency indicates the capacity of single cells to give rise to all the cells of the body, including the germ cells. In mice, three different states of pluripotency have been described that are representative of the consecutive phases of embryo development; namely, *naïve* (peri-implantation blastocyst), *rosette* (recently implanted embryo) and *primed* (post-implantation embryo) [1,2,3]. The in vitro equivalents of the pluripotency states are the embryonic stem cells (ESCs) [4], the rosette-like stem cells (RSCs) [3] and the epiblast stem cells (EpiSCs) [5,6]. It is well-known that pluripotency progresses from naïve to primed thorough a continuum of different intermediate states [7,8,9]; an exit from naïve pluripotency requires the shutdown of the naïve transcription factor network and the concomitant induction of a formative gene regulatory network [10,11,12].

Significant efforts have been made to define the culture conditions to capture intermediate pluripotency states. In addition to providing an in vitro model to investigate the mechanisms of early post-implantation development, the increasing interest is also motivated by the expectation that the intermediate states of pluripotency, unlike ESCs and EpiSCs, would be competent for the generation of primordial germ cells (PGCs), which are induced in vivo in response to BMP signaling in a defined narrow developmental time window [13]. This restricted time window as well as the few numbers of cells competent for the initiation of germ cell development in the early mouse embryo represent the major limitations to study the PGC specification in vivo. The generation of PGC-like cells from pluripotent stem cells overcomes this limitation and opens the way to investigate the molecular events segregating the germline and soma, which is a fundamental question in developmental and reproductive biology [14].

Intermediate states with transcriptional and epigenetic profiles between the naïve and primed states as well as competence for PGC differentiation have previously been described [15,16,17]. The majority of the pluripotency states isolated so far rely on the modulation of the key signaling pathways (Jak/Stat, Wnt, Fgf/Erk and Nodal/Activin) [15,17,18,19,20,21,22] through specific combinations of non-physiological chemicals (CHIR99021, IWP2, PD0325901 and XAV939) and growth factors (Lif, Fgf and Activin) (for an extensive review, see [8]). Mouse ‘poised pluripotent’ cells can also be generated by forcing the expression of the pre-mRNA splicing factor gene *ISY1* [23]; a stable human naïve-to-primed intermediate state of pluripotency has recently been isolated through lipid deprivation, which results in Erk signaling inhibition [24].

The role of metabolites and metabolic pathways in the control of the pluripotency continuum has been poorly explored until recently. One of the earliest events in the naive-to-primed transition is a metabolic reprogramming that converts bivalent naive ESCs, which use both oxidative phosphorylation (OX/PHOS) and glycolysis to produce energy, to exclusively glycolytic primed cells [25,26]. Furthermore, metabolism impacts on the differential potential of stem cells by influencing the epigenome because several intermediary metabolites act either as substrates, co-factors or products of chromatin-modifying enzymes [27]. For example, the uptake and catabolism of macronutrients as glucose or amino acids (AAs) generate substrates and/or modulators of epigenetic enzymes such as acetyl-CoA, the methyl-donor SAM and αKG. A change in the levels of these intermediate metabolites can modulate the activity of DNA and histone-modifying enzymes, thereby promoting epigenetic remodeling and facilitating differentiation (for reviews, see [28,29]). Mouse ESCs critically depend on threonine catabolism by threonine dehydrogenase (TDH) [30]. Of note, hESCs do not have THD activity and utilize methionine metabolism for self-renewal [31]. 

In this paper, we discuss the role of proline metabolism in mouse ESC identity and report on an in-depth analysis of a highly reproducible naïve-to-primed intermediate state of pluripotency that relies on the availability/level of the non-essential amino acid, proline, which was first reported almost two decades ago. 

## 2. Proline Levels Influence Pluripotency

Using different experimental approaches, two independent research groups found that proline levels influenced ESC identity. In 1999, Rathjen and colleagues first reported that a conditioned medium (MEDII) of human hepatocellular carcinoma G2 (HepG2) cells induced mouse ESC differentiation toward cells with features of an early epiblast [32]. In this seminal work, the MEDII-induced cells were named primitive ectoderm-like (EPL) cells. Later on, the same research group identified proline as the active molecule in a HepG2-conditioned medium through chromatographic fractionation and reported the role of sodium-coupled neutral amino acid transporter 2 (SNAT2) in proline transport/uptake [33,34]. EPLs differ from ESCs with respect to their colony morphology, cell cycle/proliferation rate, expression of pluripotency/differentiation markers and differentiation potential [33,35,36]. In 2011, Casalino and co-workers, using a robotic workstation for the cell-based screening of compounds libraries, identified proline and ornithine as inducers of ESC proliferation and described in detail the phenotypic transition induced by proline supplementation on ESC colonies [37]. At that time, contradictory data on the phenotypic and functional features of the cells described by the two groups led Casalino and co-workers to adopt a more neutral definition; that is, ‘proline-induced cells’ (PiCs). PiCs generate chimeric embryos upon blastocyst injections [37], suggesting that, unlike those described for EPLs [32], PiCs are developmentally proximal to naïve ESCs and may correspond with an early-primed pluripotency state. After almost a decade of studies, a few of the contradictory results might be explained—at least, in part—in terms of different genetic backgrounds and technical reasons. For instance, it is now well-known that early-primed/primed stem cells are highly susceptible to trypsin digestion; the different enzymatic dissociation used for EPLs and PiCs (i.e., trypsin vs. accutase) might explain a few discrepancies [32,37]. Cell dissociation with trypsin reduces the capacity of cells to re-establish effective cell–cell adhesive interactions, which is essential for proper cell integration in the *inner cell mass* (ICM) of the host blastocyst. Thus, it is possible to hypothesize that EPLs and PiCs represent the same naïve-to-primed intermediate state of pluripotency although direct experimental evidence is lacking.

### 2.1. Phenotypic Heterogeneity and Metastability of PiCs

Cytoskeleton structure/networks and morphology of PiCs: PiCs are obtained upon the exogenous supplementation of L-Proline to ESCs cultured in serum/Lif (FBS/Lif) on gelatin-coated dishes [37,38]. In these culture conditions, PiCs proliferate and give rise to highly irregular cell colonies showing three distinctive zones: a dome-shaped core of adherent naïve-like cells; a peripheral monolayer zone of polygonal/epithelial-like cells; and a crown of mesenchymal-like cells that detach from the colony (Figure 1, top left). Naïve-like cells are round-shaped with a prominent nucleus and a reduced cytoplasm whereas detached cells display an irregular morphology with filopodia and lamellipodia protrusions [39]. Staining with *β*-tubulin and Vinculin antibodies and with Phalloidin shows distinctive mesenchymal features such as elongated and polarized F-actin stress fibers, frequently terminating in large and mature focal adhesion complexes (Figure 1, bottom left) [39]. 

Furthermore, PiCs and ESCs exhibit different side scatter (SSC) and forward scatter (FSC) parameters by FACS, indicating that a high proline regimen (i.e., exogenously added proline) modifies the size and granularity of the cells [40]. Of relevance, the PiCs phenotype is fully reversible; i.e., upon dissociation with accutase, PiCs fully revert to naïve ESCs when cultured in an FBS/Lif medium without proline supplementation [39]. Interestingly, PiCs also revert to naïve ESCs by the addition of ascorbic acid (vitamin C; see below). This phenotypic plasticity fits with the idea that PiCs capture an intermediate state of pluripotency that can adapt to transient metabolic perturbations, in part through a reversible epigenetic mechanism (see below).

PiCs acquire a motile/invasive phenotype: Time-lapse video microscopy reveals the phenotypic ‘metastability’ of PiCs [39]. Freely motile PiCs detach from the colony edges through the extension of highly dynamic and prominent protrusions and frequently divide and re-establish cell–cell adhesive contacts with neighboring motile cells or with the core of the colony [39]. Thus, PiCs are prone to generate cell–cell and cell–substrate interactions in a highly dynamic and reversible manner. The ESC-to-PiC transition resembles key features of the partial epithelial-to-mesenchymal-like transition; it was thus named the embryonic stem cell-to-mesenchymal-like transition (esMT) and its reversion was named the mesenchymal-to-embryonic stem transition (MesT). Motile PiCs are able to invade matrigel in response to gradients of serum or different chemo-attractants, including Cyr61, Egf, insulin and Sdf-1, and were able to generate lung metastasis upon a tail vain injection in immunocompromised mice [39]. The acquisition of mesenchymal traits and the motile/invasive features of PiCs correlate with the induction of the expression of key mesenchymal markers, including *Brachyury*, *N-cadherin* and *Vimentin*, but not *Slug* and *Snail* [39]. Accordingly, the expression levels of the cell–cell adhesion protein E-cadherin (E-cad) are comparable between PiCs and ESCs; however, in motile PiCs, E-cad becomes delocalized from the plasma membrane and is mainly confined to intracellular vesicles; this was stained positive for trans-Golgi markers (Figure 1, top right) [39]. Accordingly, unlike that described in a canonical EMT and in the ESC-to-EpiSC transition, esMT is not associated with the transcriptional downregulation of the E-cad coding gene *Cdh1* [39]. In conclusion, PiCs dynamically fluctuate between two extreme phenotypes, an epithelial-like with a high tendency to preserve cell–cell interactions and a mesenchymal-like with a high cell–substrate affinity. The molecular mechanism(s) controlling the dynamic relocalization of E-cadherin during such fluctuations remains unknown and deserves further investigation. 

### 2.2. Transcriptomic and Epigenomic Landscapes of PiCs 

ESC-to-PiC transition is accompanied by extensive remodeling of the transcriptome: Proline supplementation induces the extensive remodeling of the ESC transcriptome (Figure 2, top right), inducing the differential expression of ~1500 protein-coding genes by at least 1.5-fold [39]. The expression pattern of these genes resembles in part that of FGF/Activin A (F/A)-induced EpiSCs [39,41]. In particular, PiCs and EpiSCs share ~30% of the differentially expressed genes (DEGs). Naïve pluripotency-associated genes, including *Stella*, *Pecam1*, *Klf4*, *Nanog*, *Fbx015* and *Tbx3*, are downregulated in PiCs as in EpiSCs although to a much lesser extent. Complementary to that, the priming markers *Fgf5, Brachyury, Pitx2, Otx2*, *Gata6 and Foxa2* are upregulated in PiCs although less strongly than in EpiSCs. Thus, PiCs and EpiSCs share a common set of DEGs, despite their different expression levels [39,41]. In line with the morphological and functional phenotype of PiCs, ~10% of DEGs are related to the cytoskeleton whereas ~12% correlate with cell adhesion/junction and cell motility. Moreover, ~11% of DEGs are involved in extracellular matrix (ECM) remodeling, including secreted proteases and the signaling pathways involved in cancer invasiveness such as Jak/Stat, Pi3k, MAPK and Tgfβ pathways [39,41]. Thus, the transcriptome profile provides molecular support to the morphological and functional features of PiCs.

Non-coding RNAs in the ESC-to-PiC transition: The induction of PiCs is associated with the deregulation of several classes of non-coding RNAs, including microRNAs, long non-coding RNAs (lncRNAs) and transcribed ultra-conserved elements (T-UCEs). These non-coding RNAs are emerging as key regulators of the balance between self-renewal and differentiation in pluripotent and adult stem cells [43,44]. In this context, the ultra-conserved RNA uc.170+, also named T-UCstem1, whose expression is downregulated upon exit from naïve pluripotency [45,46], is also deregulated in the ESC-to-PiC transition (Fico A.; personal communication). Whether uc.170+ has a functional role in the generation of PiCs remains to be further investigated.

The role of microRNAs in lineage progression has been extensively investigated. In particular, the expression profile of the microRNA clusters *miR-290/295* and *miR-302/367* is highly relevant in the transition from naïve to primed pluripotency [47]. The expression of the *miR-290/295* cluster marks the naïve state and is gradually downregulated as the cells exit naïve pluripotency and progress into a primed state, which, in turn, correlates with the induction of *miR-302/367.* The co-expression of *miR-290/295* and *miR-302/367* identifies an intermediate state that corresponds with an early post-implantation epiblast [47]. Although naïve ESCs almost exclusively express the *miR-290/295* cluster and F/A-induced EpiSCs mainly express the *miR-302/367* cluster, *miR-290/295* and *miR-302/367* are co-expressed in PiCs (Figure 1, bottom right), which is in line with the idea that PiCs identify an early-primed state of pluripotency [41].

Epigenome remodeling in PiCs: The naïve-to-primed transition is accompanied by a global increase in DNA and histone methylation [48]. Several lines of evidence indicate that this also occurs in the ESC-to-PiC transition. A time-course mass spectrometry analysis revealed that the global level of DNA 5mC progressively increases whereas that of 5hmC decreases in the ESC-to-PiC transitions (Figure 2, bottom left) [41]. Furthermore, a bisulfite-Seq (RRBSeq) analysis identified ~1.000 differentially methylated regions (DMRs) between ESCs and PiCs that are distributed throughout all chromosomes [41], with ~50% of them located in gene promoter regions and ~20% in gene enhancers. Of note, the majority (>90%) of hypermethylated DNA regions in PiCs were conversely hypomethylated in ESCs upon the supplementation of ascorbic acid (VitC) to the culture medium, suggesting that VitC and proline oppositely modulate methylation at the same genomic sites. It is well-known that VitC is essential for the activity of a specific class of DNA demethylases; i.e., VitC/Fe(II)/α-ketoglutarate-dependent Tet enzymes. The large majority (~90%) of hypermethylated regions in PiCs are hypermethylated as well in Tet1–3 triple knockout ESCs lacking Tet activity [49], thus providing further evidence for the idea that proline and VitC oppositely regulate the activity of Tet enzymes. Another class of epigenetic enzymes that similarly require VitC for their activity is the Jmjd family of histone demethylases. The PiC specification is accompanied by a significant increase in histone H3 lysine 9 (H3K9) and histone H3 lysine 36 (H3K36) methylation levels (Figure 2**,** bottom right) [39]. In PiCs, H3K9me3 methylation is significantly altered at ~16.000 sites, mostly located in non-coding intergenic regions, and in ~30% of the ~1.500 DEGs [39]. Conversely, H3K9me3 increases in pericentromeres and gene deserts, suggesting heterochromatin reorganization toward more dense nuclear structures [39]. Of note, chromatin remodelers are released from translation inhibition by an RNA-induced silencing complex (RISC) and play a key role in the naïve-to-primed transition [50]. H3K36me3 methylation is also significantly altered in ~8.000 sites in PiCs, with a large fraction (~40%) of PiC-specific genes showing significant H3K36me3 changes according to the transcriptional status [39]. Together, these findings lead us to propose that proline supplementation interferes, either indirectly or directly, with the activity of VitC/Fe(II)/α-ketoglutarate-dependent epigenetic enzymes and this eventually results in an increase in DNA and histone methylation that accompanies the ESC-to-PiC transition (Figure 3).

Several lines of evidence indicate that changes in the abundance of different metabolites could modify the epigenome in a reversible manner [27]. Similarly, proline supplementation may reduce the availability of the metabolites/co-factors required for the activity of Tet and Jmjd enzymes, including VitC (see above; Figure 3). It will be interesting to investigate in the future whether other epigenetic enzymes may be involved in this process, including histone acetyl transferases (HATS) and histone deacetylases (HDAC), which require acetyl-CoA [24].

### 2.3. Amino Acid Starvation Stress and Energy Metabolism in the ESC-to-PiC Transition

Proline transport and intracellular fate: To induce proliferation and the ESC-to-PiC transition, exogenously added proline must be internalized/transported into the cell cytoplasm. Tan and co-workers proposed a critical role for the Slc38a2 amino acid transport system (also known as Snat2) in PiC specifications [34]. As expected, the co-supplementation of proline and SLC38a2 substrates such as serine and alanine reduced both the proline uptake and the ESC-to-PiC transition [33,37]. NMR-based metabolomic profiling has revealed variations in specific amino acid levels in PiCs vs. ESCs (Figure 2, top left). Specifically, whilst the levels of proline and glutamic acid increased, those of alanine, glutamine, glycine, leucine, serine, tyrosine and valine decreased. The relative levels of intracellular proline in ESCs, PiCs and embryonic fibroblasts were 1, 4 and 10, respectively [42]. Of note, intracellular proline can be either used to load free prolyl-tRNA and increase translation (mainly of proline-rich proteins, including collagens) or it can be channeled from the cytoplasm into the mitochondria, where it is catabolized to obtain energy/ATP and glutamate, which is a trait of human cancer cells [51,52]. It is known that mitochondria proline catabolism generates reactive oxygen species (ROS), which have been implicated in hypersensitive responses in different cellular contexts from plants to humans [53,54,55]. Conversely, there is evidence to suggest that proline-induced ESC proliferation and AAR inactivation do not correlate with ROS accumulation ([42] and D’Aniello et al., personal communication). Furthermore, a metabolic analysis of PiCs also proved that proline supplementation did not exert any oxidizing effect and even increased the levels of reduced glutathione (GSH) compared with ESCs [41]. However, direct evidence based on PRODH-mutant ESCs is still lacking.

Alleviation of amino acid starvation stress: One of the earliest molecular events that occurs after proline uptake is the downregulation of the transcription factor Atf4 and its target genes, including *Asns*, *Trib3* and *Nupr1* [42]. Atf4 is the downstream effector of the amino acid starvation response (AAR) pathway that is activated by uncharged tRNAs that activate general control nonrepressed 2 (Gcn2) protein kinase, leading to the phosphorylation of Eukaryotic initiation factor 2 (Eif2α) and Atf4 translation. Consistently, Eif2α was rapidly dephosphorylated upon proline addition to a culture medium [42]. Several results support the idea that proline induces the alleviation of ARR stress, including: (i) the demonstration that Halofuginone, a specific inhibitor of Prolyl-tRNA synthetase (Prs) [56], is a potent inhibitor of the PiC specification [42]; and (ii) the opposite effects of Atf4 overexpression (Tet-off Atf4 ESCs) and downregulation (ShRNA-Atf4 knockdown) on the ESC-to-PiC transition [42]. Moreover, Atf4 expression/translation is regulated by proline levels (high proline levels/low Atf4) and proline levels are, in turn, controlled by Atf4 through proline biosynthetic enzymes Aldh18a1 (P5C synthetase) and Pycr1 (P5C reductase), which are under the transcriptional control of Atf4 [42]. Thus, under proline-limiting conditions, uncharged tRNAs accumulate and activate/phosphorylate both Gcn2 and Eif2α, inducing the expression of Atf4, which, in turn, activates the expression of the genes involved in proline uptake (*Slc38a2*) and proline biosynthesis (*Aldh18a1* and *Pycr1*). This autoregulatory loop generates a partial starvation of proline that induces a compensative AAR stress pathway and preserves the ESC identity [42]. This is further supported by the finding that the autophagy pathway is active in FBS/Lif ESCs and it is rapidly repressed after proline supplementation [42].

Energy metabolism in PiCs: Global metabolomic profiling using mass spectrometry revealed significant differences in energy-related metabolites between PiCs and ESCs [41]. In particular, PiCs were more reliant on the glycolytic pathway to obtain energy/ATP compared with ESCs. PiCs produce more lactate and are more susceptible to the inhibitors of glycolysis such as 2-deoxy-D-glucose (2-DG). Moreover, PiCs show a significant reduction in the potential of the mitochondrial membrane and thus in the activity of the oxidative phosphorylation pathway [41]. Therefore, similar to observations in EpiSCs [25], PiCs are glycolytic-dependent cells. These findings indicate that PiCs are highly glycolytic, which is particularly relevant as the earliest event in the transition from the naïve to the primed state is a metabolic switch from oxidative phosphorylation to glycolysis [25,26].

The mechanism of the ESC-to-PiC transition and the role of collagen synthesis/hydroxylation: Proline availability influences the loading rate of prolyl-tRNA and consequently the expression/translation of proline-rich proteins such as collagens. Collagens are then subjected to different post-translational modifications, including the hydroxylation of proline residues by ER-residing enzymes, collagen Prolyl 4-hydroxylases (P4hs). As with Tets and Jmjs, P4hs belong to the family of VitC/Fe(II)/α-ketoglutarate-dependent enzymes, which require VitC (Figure 3, top right). We thus propose that a sudden increase in the collagen synthesis provokes a compartmentalized metabolic imbalance that eventually alters the epigenetic landscape of the stem cells. Briefly, the P4h-dependent hydroxylation of nascent collagens in the ER consumes α-ketoglutarate and VitC, which are also essential substrates/co-factors for the activity of the nuclear hydroxylases/dioxygenases Tet and Jmj, which, in turn, catalyze DNA and histone demethylation, respectively. Genetic and pharmacological evidence indicates that the ESC-to-PiC transition is strictly dependent on P4h-dependent collagen hydroxylation and it is counteracted by a high VitC regimen, which prevents proline-induced epigenetic changes and the ESC-to-PiC transition [57]. We thus propose a model whereby the identity of ESCs relies, in part, on a functional metabolic interplay between collagen and DNA/histone, whose imbalance triggers the exit of ESCs from naïve pluripotency, inducing lineage commitment [27]. A similar metabolic interplay occurs in cancer cells [57].

### 2.4. Pluripotency Features of PiCs

PiCs retain pluripotency in vitro and in vivo: PiCs are able to differentiate into cardiac and neural cells in vitro under specific culture conditions. Specifically, PiC-derived embryoid bodies (EBs) generate areas of contracting cardiomyocytes [37]. The differentiation into cardiac muscle cells was confirmed by molecular and immunofluorescence analyses of different cardiac markers, including Nkx2.5 and α-myosin heavy chain (α-MHC) [35,37]. Moreover, under 2D culture conditions [58], PiCs spontaneously differentiated into neuronal and glia cells (Figure 4, bottom right) [37,59]. However, PiC-derived 3D EBs do not efficiently differentiate into neurons, unless retinoic acid is added to the culture [35].

PiCs also display features of pluripotency in vivo. PiCs were able to generate teratomas comprising tissue representatives of all three embryonic germ layers after a subcutaneous injection in the flank of immunocompromised mice (Figure 4, bottom left) [37]. Moreover, Casalino and co-workers showed that eGFP-labelled PiCs were able to contribute to chimeric mouse embryos (E11.5) upon an injection into the host blastocyst with high efficiency (Figure 4, top left) [37]. Conversely, Rathejn and co-workers reported that MEDII-induced EPLs, unlike ESCs, were unable to generate chimeric mouse embryos [32]. This inconsistency might be explained—at least, in part—by the different cell dissociation methods (see above, Proline levels influence pluripotency).

PiCs generate elongated gastruloids: Recently, the efficiency to generate 3D embryo-like structures named gastruloids has been proposed as a method to evaluate the developmental fidelity of stem cells [40]. Gastruloids are 3D aggregates of ESCs that, under defined conditions, display an axial organization and induction of the hallmarks of post-implantation development [60]. The efficiency of gastruloid formation decreases as pluripotency progresses from the naïve to the primed state. Naïve ESCs show a very high gastruloid formation efficiency (>90%) and transient early-primed EpiLC aggregate, but then fail to elongate whereas primed EpiSCs do not even aggregate [40]. Conversely, PiCs (250–300 cells) are able to spontaneously aggregate and generate spheroids, which elongate and properly express the developmental genes along the anterior–posterior (A–P) axis although with a lower efficiency compared with that of naïve ESCs (Figure 4, top middle) [40]. PiC aggregates show a premature extension and are smaller in size compared with naïve ESC-derived gastruloids. In line with these findings, Lake and co-workers showed a precocious differentiation of mesodermal precursor cells in PiCs compared with naïve ESC-derived EBs [35]. Thus, unlike EpiLCs, PiCs identify an intermediate state of pluripotency that maintains a competence to generate gastruloid-like organoids.

PiCs acquire competence to differentiate into primordial germ cell-like cells: As with transient EpiLCs and unlike ESCs and EpiSCs [8], PiCs are competent at primordial germ cell-like cell (PGCLC) differentiation [61] (Figure 4, bottom middle). The PGCLC differentiation of PiC-derived aggregates in a GK15 medium supplemented with Lif, Bmp4, Egf and Scf was confirmed by combining qPCR and immunofluorescence analyses of the expression of PGC markers as Blimp1, Prdm14, Nanos3 and AP2-γ [61].

## 3. Discussion

Proline is a non-essential amino acid that is important for many different cellular functions from bacteria to human cells [51]. As described herein, the exogenous supplementation of proline to serum/LIF ESCs drives naïve ESCs toward a stable and highly reproducible naïve-to-primed intermediate state of pluripotency with a competence for a primordial germ cell-like (PGCL) fate. In the last decade, significant efforts have been made to characterize this cell state at molecular, metabolic and functional levels. A detailed protocol to generate proline-induced cells, named EPL, has recently been reported [38].

The exit from naïve pluripotency in vivo is accompanied by a substantial increase in molecular and phenotypic cell heterogeneity. Cellular heterogeneity, in terms of gene expression and morphology, was observed since the formation of the ICM in mouse blastocysts [62,63,64]. Similar changes accompany the transition from naïve ESCs to PiCs as well as the generation of other intermediate states of pluripotency, including RSCs [3] and intermediate epiblast stem cells (IESCs) [19]. More dynamic, rather than static, models of ESC early differentiation are thus preferred to mimic the in vivo developmental process [65]. The dynamic heterogeneity of the PiC population and the use of physiological doses of a natural amino acid to induce this pluripotent state nicely approximate the natural process of early differentiation priming and do not rely on a combination of non-physiological chemical inhibitors and/or growth factors.

PiCs have a transcriptional profile of cells that exit the naïve state and enter the primed state. A PCA analysis of transcriptome data placed PiCs between naïve ESCs and primed EpiSCs [41]. PiCs and EpiSCs share a large percentage of common genes with the same deregulation trend, but with a different magnitude. Furthermore, PiCs rely on the activation of endogenous Fgf and Tgfβ pathways and are capable of responding to bFgf and Activin A [41].

The ESC-to-PiC transition is accompanied by significant chromatin remodeling. Thus, proline adds to the increasing list of metabolites that have been described so far, to the best of our knowledge (threonine, butyrate, α-ketoglutarate and ascorbic acid) that act as epigenetic modulators of ESCs and/or somatic cell reprogramming [66,67,68]. Specifically, proline influences the global level of the trimethylation of lysines K9 and K36, inducing a global and genome-wide increase in H3K9 and H3K36 trimethylation, preferentially at non-coding intergenic regions and constitutive heterochromatin [39]. In line with these findings, the silencing of H3K9 demethylase Jmjd1a induces PiC-like features, including a sensitivity to trypsin digestion, the formation of irregular flat-shaped colonies and the upregulation of *Fgf5* and *Brachyury* [69]. The ESC-to-PiC transition is also associated with a global and genome-wide increase in DNA methylation levels that is counteracted by vitamin C. Changes in chromatin remodeling and increased DNA methylation levels are known to be associated with the induction of primed pluripotency both in vivo and in vitro. Accordingly, proline and VitC oppositely modify DNA methylation at genomic regions that normally gain methylation during the blastocyst to epiblast transition [41,70].

The mechanisms by which proline acts as an epigenetic signal have recently been suggested to rely on a previously unexplored collagen–epigenetic axis [57]. That the intracellular level of a non-essential metabolite such as proline in naïve ESCs is limiting might appear to be a paradox. However, this is easily explained because proline limitation is essential to prevent the exit from naïve pluripotency and to maintain the ESC identity [42]. To achieve this, proline biosynthesis is under the control of the AAR–Atf4 pathway that is active in naïve FBS/Lif ESCs. Proline supplementation to naïve FBS/Lif ESCs inactivates the AAR–Atf4 stress pathway, induces collagen synthesis and reduces mitochondrial oxidative phosphorylation [41,42,57]. Similar changes in energy metabolism (bivalent vs. glycolytic) occur in the ESC-to-EpiSC transition [5,6]. Although proline downregulates the expression of the OXPHOS enzymatic complex, a putative inhibitory effect of mitochondrial proline oxidation, i.e., the conversion of proline into glutamate [51,71], is not excluded.

In line with the idea that PiCs resemble an early-primed state of pluripotency, PiCs are able: (i) to differentiate into derivatives of the three germ layers in vitro and in vivo; (ii) to integrate into the blastocyst and contribute to chimeric embryos; and (iii) to differentiate into primordial germ cell-like cells (PGCLCs), which is a unique feature of pre-gastrulation epiblast-like cells [61]. Furthermore, unlike EpiLCs and EpiSCs, PiCs are able to generate elongated gastruloids [40,72]. Interestingly, embryoid bodies (EBs), gastruloids and PGC differentiation all rely on the formation of floating cellular aggregates. Although naïve ESC-derived aggregates appear to be highly homogenous, round-shaped and compacted, PiCs generate loosely packed aggregates that are more irregular and disorganized [35,40]. Although a further analysis is needed, this phenotype may be explained, in part, by the heterogeneity of PiCs; only a fraction/subpopulation of PiCs might be able to engage in stable cell–cell interactions based on a cytoskeleton organization [73] and/or cell surface dynamics/fluctuations [63].

In conclusion, PiCs define a stable intermediate state of pluripotency that shows features of an early-primed state of pluripotency. We believe that this unique dynamic model of ESC differentiation may provide insights into the complexity of the early phase of the developmental process.

## Figures and Tables

**Figure 1 cells-11-02125-f001:**
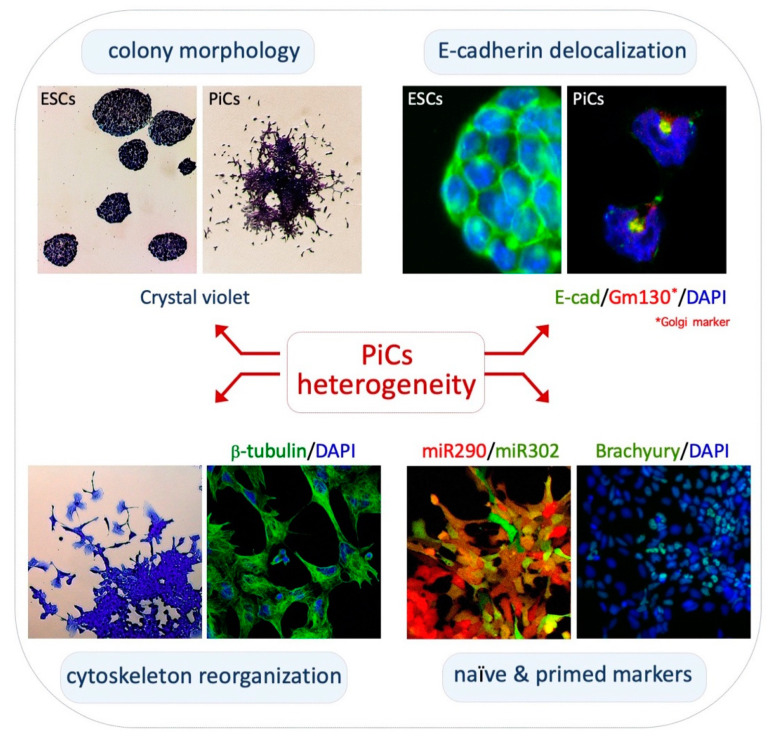
Phenotypic heterogeneity of PiCs. **Top left**, crystal violet staining highlights the irregular/heterogenous structure of PiC colonies. **Top right**, immunofluorescence analysis showing the different localization (cell membrane vs. perinuclear structure) of E-cadherin protein (green) in naïve ESCs and motile PiCs. **Bottom left**, motile PiCs show a mesenchymal-like morphology with prominent filopodia and lamellipodia and a fibrillar cytoskeleton. **Bottom right**, expression of miR-290/mCherry and miR-302/eGFP (**left**) in PiCs derived from dual reporter ESCs (DRES) and immunostaining for Brachyury (**right**) in PiC colonies.

**Figure 2 cells-11-02125-f002:**
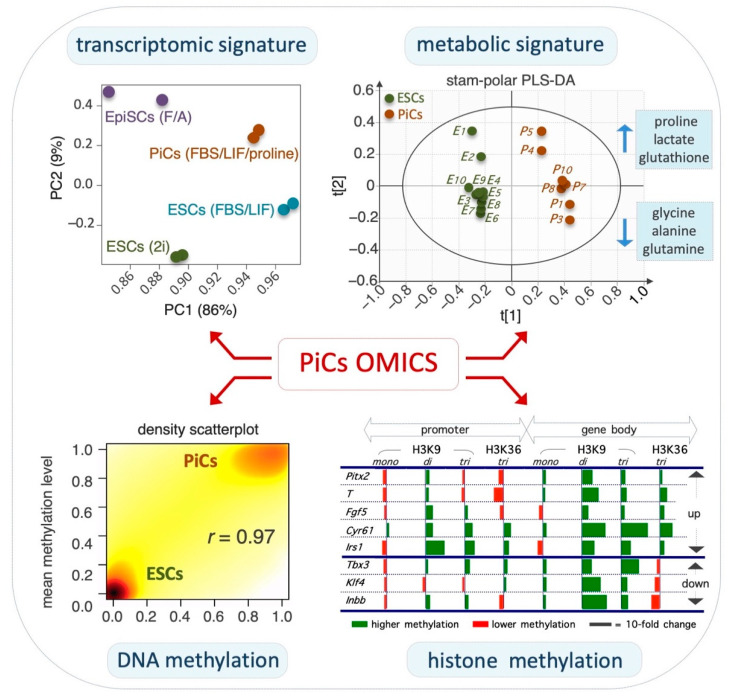
Global omics associated with PiC specifications. **Top left**, principal component analysis of the transcriptome showing that PiCs display a gene expression profile between naïve ESCs and primed EpiSCs. **Top right**, scores plot of the polar fraction of cell metabolome clearly discriminates PiCs from ESCs. **Bottom left**, scatterplot showing that the global DNA methylation level is higher in PiCs compared with ESCs. **Bottom right**, the relative level (PiCs vs. ESC, fold-change) of H3K9 and H3K36 histone methylation marks in selected genes. Data are modified and adapted from [39,42].

**Figure 3 cells-11-02125-f003:**
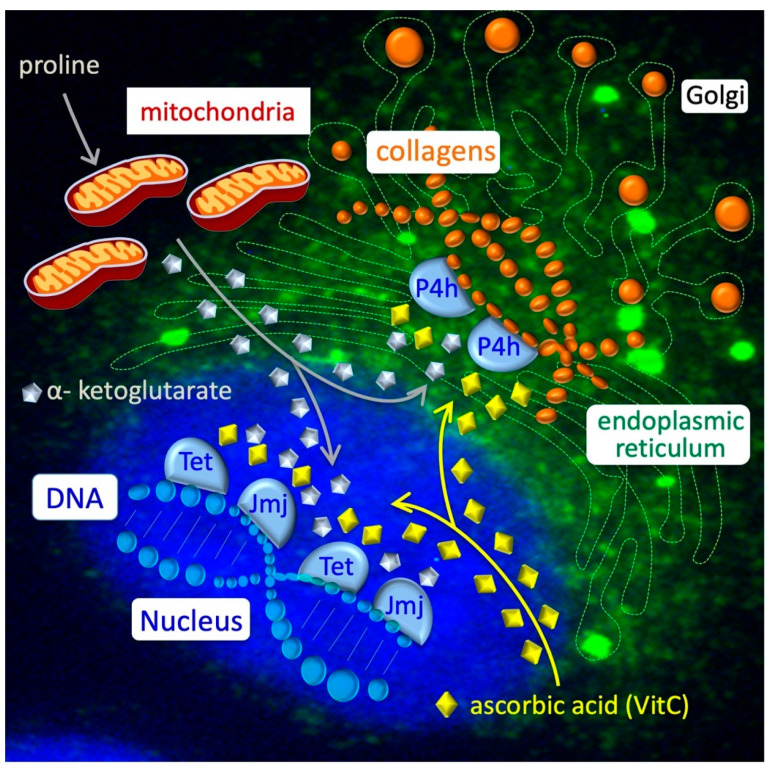
Collagen–epigenetic metabolic interplay in ESC-to-PiC transition. Intracellular proline is transported inside the mitochondria (grey line) and converted into α-ketoglutarate or used to load free prolyl-tRNA, which is essential for the synthesis of proline-rich proteins as collagens. Nascent collagens (orange balls) are hydroxylated in proline residues by Prolyl 4-hydroxylase (P4h) enzymes in the endoplasmic reticulum (green) and secreted through Golgi vesicles into the extracellular space. In the nucleus (blue), Tet and Jmj enzymes catalyze hydroxylation (demethylation) of DNA and histones. Reduced ascorbic acid/vitamin C (yellow diamonds) is essential for the activity of a large family of dioxygenases, including P4h, Tet and Jmj enzymes.

**Figure 4 cells-11-02125-f004:**
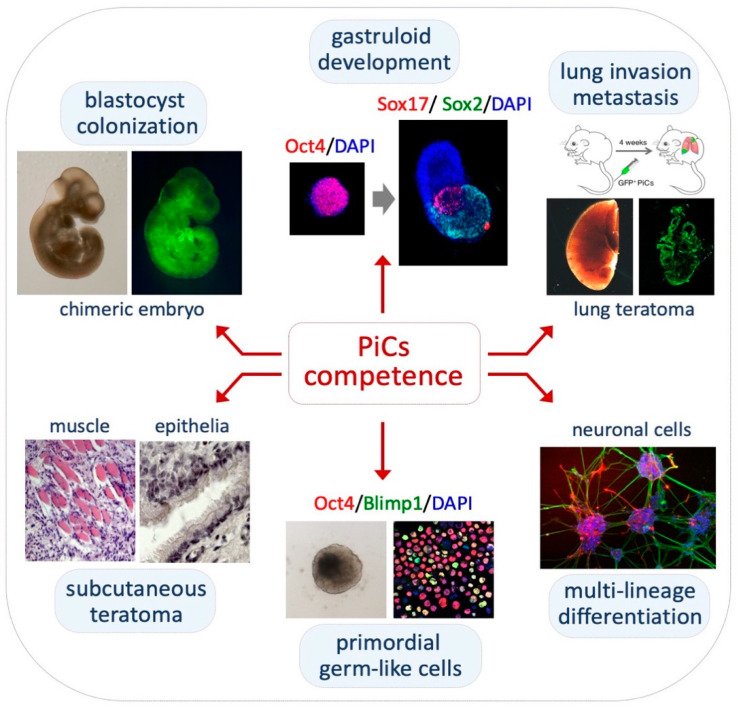
PiC pluripotency in vitro and in vivo. **Top left**, GFP-positive chimeric embryo (E11.5) generated after the injection of eGFP-labelled PiCs into the blastocyst. **Top middle**, immunofluorescence staining of *Oct4* (red) in PIC-derived spherical aggregates (**left**) and Sox17 (red) and Sox2 (green) in elongated gastruloids. **Top right**, GFP-positive lung tumor generated after tail vein injection of eGFP-labelled PiCs. **Bottom left**, hematoxylin–eosin-stained sections of PiC-derived subcutaneous teratomas. **Bottom middle**, bright-field image (**left**) of PiC-derived aggregates and immunofluorescence staining (right) of Oct4 (red) and Blimp1 (green) on cytospin-fixed cells from PiC-derived aggregates showing the presence of Oct4/Blimp1 double-positive PGCLCs. **Bottom right**, neural lineage differentiation of PiCs.

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
