# Peer review of "Capturing Transitional Pluripotency through Proline Metabolism"

_cells, 2022, doi:10.3390/cells11142125_

Round 1
Reviewer 1 Report
The manuscript (cells-1789170 ) “Capturing transitional pluripotency through proline metabolism” from Minchiotti et al., is an interesting review on the role of proline metabolism in the control of Embryonic Stem Cells (ESCs) identity. The review is well written and organized and captures well the current knowledge on this topic.
I do not have major criticisms, but only a few suggestions and criticisms that hopefully might further improve the quality of this manuscript
Minor concerns
Line 38. Define the acronym BMP ( at its first occurrence or, at least, in the abbreviations)
Line 34-39. I suggest spending a few words about primordial germ cells (PGCs) and explaining why is so important to have pluripotent cells able to give rise to PGC cells.
Line 47. gene ISY1, should be written in italics
Line 73-87. The difference between EPLs and PiCS remains largely unexplained, and the hypothesis that the differences mainly rely on technical problems (i.e. trypsin vs accutase) is plausible but not experimentally proved. Accordingly, I would be more cautious in saying that EPLs and PiCs represent the same naïve-to-primed intermediate state of pluripotency.
Line 119. I would suggest changing “ they frequently divide” to “and frequently divide”
Line 146. Correct "these gene" in "these genes"
Line 148. Put and hyphen between “pluripotency” and “associated” (pluripotency-associated genes).
Line 151. Put a comma between " PICs" and "although"
Line 152. "share" not "shares"
Line 154. "correlate" not "correlates"
Line 169. Consider inserting a comma to separate long non-coding RNAs (lncRNAs), and transcribed ultraconserved elements (T-UCEs).
Figure 2. Top right Stam-polar PLS-DA. Is "stam" correct?
Line 211. Remove "of" at the end of this line
Line 213. I suggest changing "PiCs specific genes" in "PiC-specific genes"
Line 229. Change "exogenous" in "exogenously"
Line 241-244. I have no doubts that D’aniello et al. ( ref. 34) found no correlation between exogenous proline and redox balance. However, other authors reported different observations (see, for example, Liang et al. ANTIOXIDANTS & REDOX SIGNALING (2013) 19, or Ferreira et al. Neurotox. Res. 2021 39(2):327-334). Given the nearly universally recognized importance of proline catabolism and the complexity of proline metabolism on cell behavior, I would be more cautious in expressing clear-cut conclusions or credit alternative hypotheses from author authors. I would be more cautious in expressing clear-cut conclusions. Incidentally, this observation highlights a slight tendency to self-referring ( 17 self-references out of 64 total references= 26.5%), although this is not necessarily a problem.
Line 258. Put “of “ before the last “the” (Atf4 is the downstream effector of the amino acid starvation response)
Line 363. The sentence “a high proline regimen drives” could deserve in discussion some further consideration. Several authors have shown that different intracellular levels of proline may have very different effects, with low levels usually associated with positive effects and high levels usually associated with negative effects, at least under normal conditions. Accordingly, I suggest defining more precisely “low” and “high” proline, and possibly discussing this issue.
Reviewer 2 Report
The current review by Minchiotti et al., “Capturing transitional pluripotency through proline metabolism” is well written and has summarized the proline-induced primed ESCs’ intermediate state of pluripotency. The authors highlighted how PiCs are distinct from ESCs at morphological, transcriptional and levels. The authors as well described the global shift H3K9 and H3K36 trimethylation in PiCs. Overall the review provided a comprehensive overview on the significance of Proline in naïve-to-prime intermediate state of pluripotency. However, there are few minor comments, which should be addressed before it is accepted for publication.
Minor comments:
1. In the introduction, the authors briefly discuss about metabolic switch and describe the metabolic pathways that stem cells adapt during activation and differentiation, underlining the significance metabolites.
2. Also, before jumping on to proline in the last paragraph of intro, the authors should describe the significance and differential use of other amino acids including threonine, methionine, glutamine etc. ( A graphical representation of this in the figure 1 as 1A would be beneficial)
3. In line 116, the authors should add a statement on the inference/or their perspective on the reversible phenotype of PiCs.
4. For Figure 2, the authors should mention the omics data as modified and adapted from their original research article (10.1016/j.stemcr.2016.11.011)
5. Line 170, add reference after T-UCEs.
